# An Ultrasonic Guided Wave Mode Selection and Excitation Method in Rail Defect Detection

**Hongmei Shi [1,2], Lu Zhuang [1], Xining Xu [1,2,*], Zujun Yu [1,2] and Liqiang Zhu [1,2]** 

[1]  School of Mechanical, Electronic, and Control Engineering, Beijing Jiaotong University, Beijing 100044, China; hmshi@bjtu.edu.cn (H.S.); 17121287@bjtu.edu.cn (L.Z.); zjyu@bjtu.edu.cn (Z.Y.); lqzhu@bjtu.edu.cn (L.Z.)

[2]  Key Laboratory of Vehicle Advanced Manufacturing, Measuring and Control Technology, Beijing Jiaotong University, Ministry of Education, Beijing 100044, China

*  Correspondence: xnxu@bjtu.edu.cn

**Abstract:** Different guided wave mode has different sensitivity to the defects of rail head, rail web and rail base in the detection of rail defects using ultrasonic guided wave. A novel guided wave mode selection and excitation method is proposed, which is effective for detection and positioning of the three parts of rail defects. Firstly, the mode shape data in a CHN60 rail is obtained at the frequency of 35 kHz based on SAFE method. The guided wave modes are selected, combining the strain energy distribution diagrams with the phase velocity dispersion curves of modes, which are sensitive to the defects of the rail head, rail web and rail base. Then, the optimal excitation direction and excitation node of the modes are calculated with the mode shape matrix. Phase control and time delay technology are employed to achieve the expected modes enhancement and interferential modes suppression. Finally, ANSYS is used to excite the specific modes and detect defects in different rail parts to validate the proposed methods. The results show that the expected modes are well acquired. The selected specific modes are sensitive to the defects of different positions and the positioning error is small enough for the maintenance staff to accept.

**Keywords:** ultrasonic guided waves; SAFE; rail defect detection; mode excitation

## 1. Introduction

It is of great significance to discover the internal defects of rails in time for maintaining and ensuring the safety of trains. At present, the hand-push detection vehicle and large-scale detection vehicle are the main tools to detect the defect detection of Continuous Welded Rails [1–3]. These two kinds of detection vehicles are running in the maintenance time, generally from 12 a.m. to 4 a.m., and cannot be used for real-time monitoring of rail internal defects. Guided waves can propagate a long distance in the rail [4–6]. Based on the pulse-echo method [7–9], the on-line monitoring of rail internal defects can be implemented by detecting the echo signals generated at the rail defects during the propagation of the guided waves.

In the research of rail defect detection based on guided waves, some scholars selected the suitable modes for rail defect detection by a mode classification method. Hayashi [10] divided the modes of the rail base into three categories. By analyzing the dispersion characteristics of guided waves, the concept of dominant modes is proposed. In the frequency range of 60~200 kHz, the transverse and vertical vibration modes with smooth dispersion curves of rail base are more suitable for defect detection than the longitudinal vibration modes [11]. A rail base defect of 20 mm in length was successfully detected by exciting the vertical modes. Subsequently, Sheng Huaji [12] obtained the optimal excitation frequency and pulse period of the propagating modes at the rail base through simulation, and tested oblique cracks at different angles by numerical simulation. The results show

that the detection effect of the vertical vibration modes is better than the transverse vibration modes. Lu Chao et al. [13] excited the vertical vibration modes on the rail head with a kind of mode force hammer. The experimental results show that the vertical vibration modes can effectively detect the transverse crack of the rail head. In addition, some scholars chose the modes to detect the damage of different parts of the rail based on the mode energy distribution. Gharaibeh et al. [14] selected the mode with energy concentrated on the rail head to detect the rail head defect. And the signal was excited according to the experimental method. The transverse defect of rail head at a distance of 9 m from the excitation location was successfully identified. In order to distinguish large transverse cracks with a diameter of 30 mm from 6 mm welded seam, Long and Loveday [15] compared the symmetric mode 1 and the antisymmetric mode 2 focusing energy at the rail head, and the symmetric mode 1 successfully distinguished the crack and welded seam at the rail head. Mode 3 and mode 4 focused with energy at the rail web, mode 4 was suitable for detecting the welded seam as well as damage in the rail web. It can be seen that in the research of the excitation mode applied to the detection of rail defects, most of the scholars use the experimental methods to excite the expected modes without specific theoretical guidance and mathematical model. The excitation signal usually contains multiple modes, which also makes it difficult for echo signal analysis and defect location.

In view of the above problems, this paper will focus on the guided wave mode selection and excitation methods based on a mathematical model in rail defect detection, by which the modes can be selected and excited with different sensitivities to the defects of the rail head, rail web and rail base, and accordingly the defect position can be rapidly found in different parts of the rail. The paper is organized as follows: A mode selection method is described in Section 1, which is used to select the modes having the concentrated strain energy and small attenuation corresponded to the rail head, rail web and rail base. The mode excitation method is presented in Section 2, to acquire the optimum excitation direction and excitation node of guided wave modes. Section 3 introduces the results of the target modes excitation simulation and verification. The simulation and detection results of the rail defects are demonstrated and analyzed in Section 4.

## 2. Mode Selection Based on Semi-Analytical Finite Element (SAFE) Method

In order to study the mode selection method of guided waves in rail defect detection, it is necessary to obtain the mode shape data. Many studies have proved that SAFE method is suitable to solve the propagation characteristics of guided waves in waveguide, such as plates [16], cylinders [17], rails [18], etc. SAFE method is also used in this paper to solve the mode shape data of CHN60 rail, which is a 60 kg/m China rail. The CHN60 rail coordinate system is shown in Figure 1.

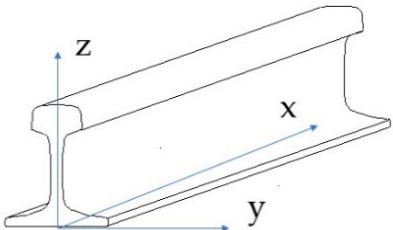

**Figure 1.** Coordinate system of a rail.

The displacement $u$, stress $\sigma$ and strain $\varepsilon$ of each particle in the rail can be expressed as Equation (1):

$$
\begin{aligned}
\boldsymbol{u} &= \begin{bmatrix} u_x & u_y & u_z \end{bmatrix}^T \\
\boldsymbol{\sigma} &= \begin{bmatrix} \sigma_x & \sigma_y & \sigma_z & \sigma_{yz} & \sigma_{xz} & \sigma_{xy} \end{bmatrix}^T \\
\boldsymbol{\varepsilon} &= \begin{bmatrix} \varepsilon_x & \varepsilon_y & \varepsilon_z & \gamma_{yz} & \gamma_{xz} & \gamma_{xy} \end{bmatrix}^T
\end{aligned}
\tag{1}
$$

Based on SAFE theory, it is assumed that the displacement field of the guided wave propagation direction is harmonic [19]. The cross section of the rail is discretized into 255 triangular elements and 177 nodes, as shown in Figure 2.

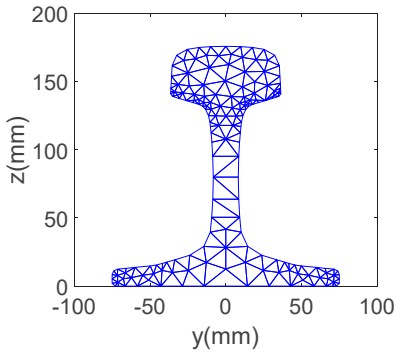

**Figure 2.** Discretization of triangle elements.

After discretization, the strain energy $\phi$ of each triangular element can be written as Equation (2):

$$\phi = \int_{V_e} \delta(\varepsilon^{(e)T})C\varepsilon^{(e)}dV_e = \delta q^{(e)^T}(\boldsymbol{K}_1 - i\xi\boldsymbol{K}_2 + \xi^2\boldsymbol{K}_3)q^{(e)} \tag{2}$$

where $q^{(e)}$ is the nodal unknown displacement of each triangular element; $\xi$ is the wavenumber; $\boldsymbol{K}_1$, $\boldsymbol{K}_2$, and $\boldsymbol{K}_3$ are stiffness matrices; $C$ is the elastic constant matrix of the rail.

By substituting strain energy and potential energy at any node in the rail cross section into Hamilton's formula, the general homogeneous wave equation of guided waves can be derived as Equation (3) [19]:

$$\left[\boldsymbol{K}_1 + i\xi\boldsymbol{K}_2 + \xi^2\boldsymbol{K}_3 - \omega^2\boldsymbol{M}\right]_M \boldsymbol{U} = 0 \tag{3}$$

where $\boldsymbol{M}$ is the mass matrix; $\boldsymbol{U}$ is the nodal displacements; $\omega$ is frequency.

By solving the eigenproblem of Equation (3), the corresponding wavenumber $\xi$ can be obtained for a given frequency $\omega$. The triangular element nodal displacements $\boldsymbol{U}$ are calculated from the eigenvalue problem. And the strain fields are reconstructed from Equation (2).

At the frequency of 35 kHz, the nodal displacement $q^{(e)}$, stiffness matrix $\boldsymbol{K}_1$, $\boldsymbol{K}_2$, and $\boldsymbol{K}_3$, and wavenumber $\xi$ are substituted into the Equation (2), and the strain energy of each mode in each triangular element can be acquired. The strain energy distribution diagram of each mode is normalized as shown in Figure 3.

At the frequency of 35 kHz, there are 20 kinds of guided wave modes propagating in the rail. Our expected mode is that the strain energy is concentrated in a single part of the rail. From Figure 3, we can see that mode 1, mode 2, mode 3, mode 7, mode 10, and mode 20 have such characteristics while strain energy of the other modes is distributed at three rail parts.

In Figure 4a,b, the strain energy of mode 7 and mode 20 is concentrated on the rail head. In Figure 4c, it can be seen that the strain energy of mode 3 is focused on the rail web. In Figure 4d, Figure 4e,f, the strain energy of mode 1, mode 2, and mode 10 is concentrated on the rail base. Therefore, mode 7 and mode 20 can be selected to detect rail head defects, similarly, mode 3 for rail web and mode 1, mode 2 and mode 10 for rail base.

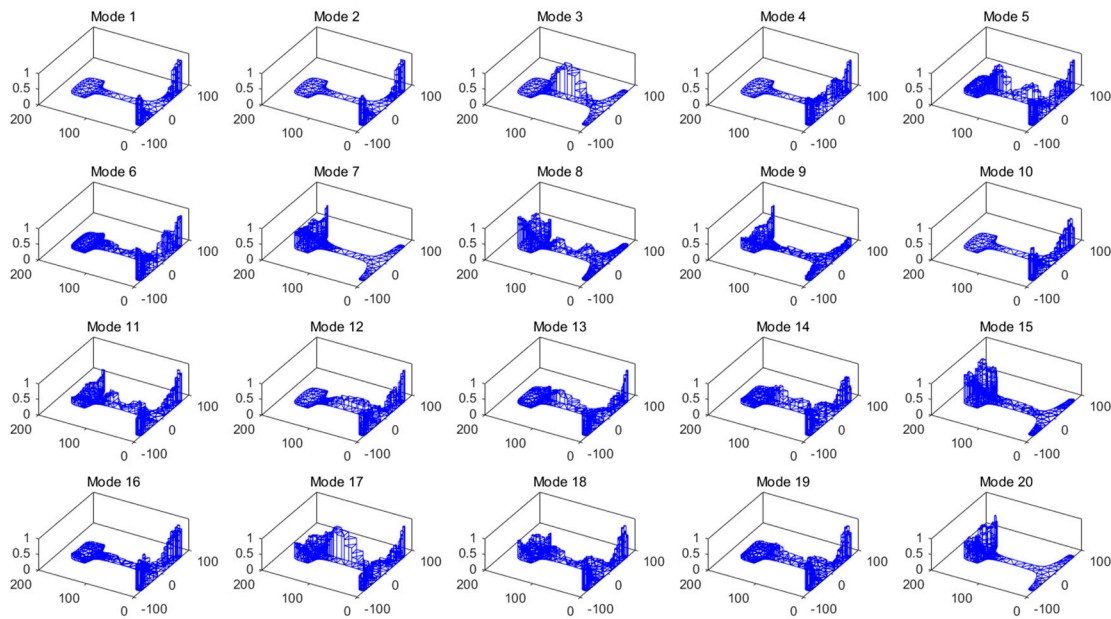

**Figure 3.** Strain energy distribution diagram of modes.

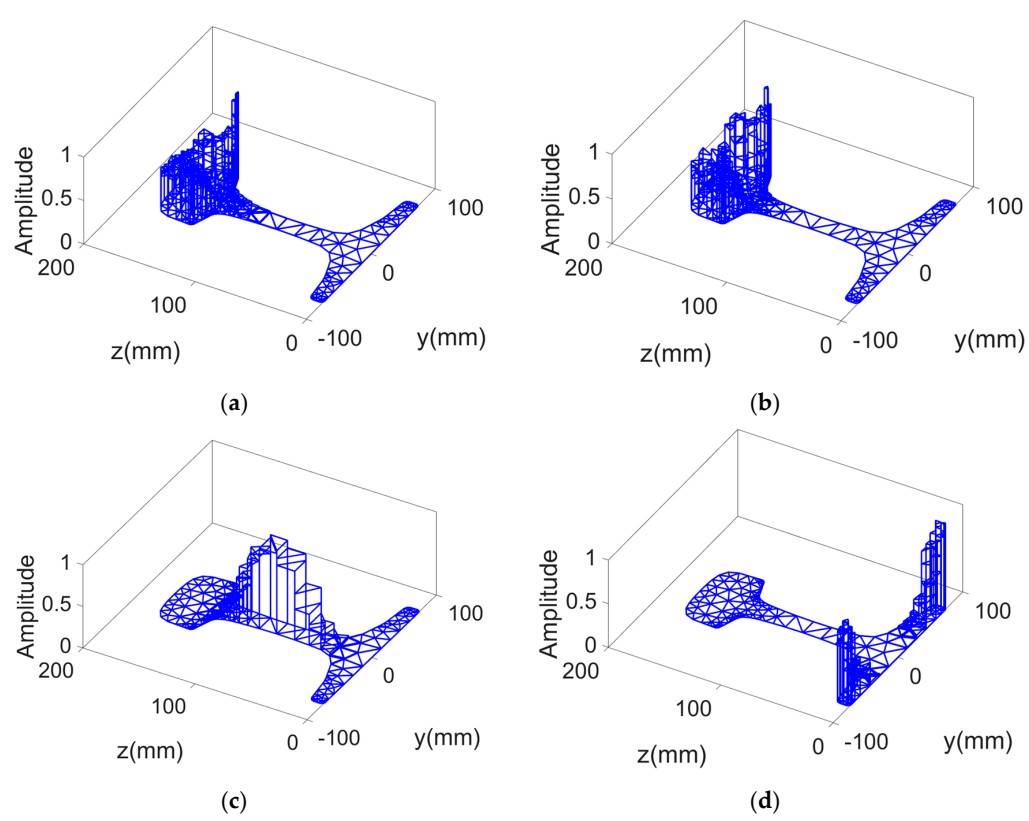

**Figure 4.** *Cont.*

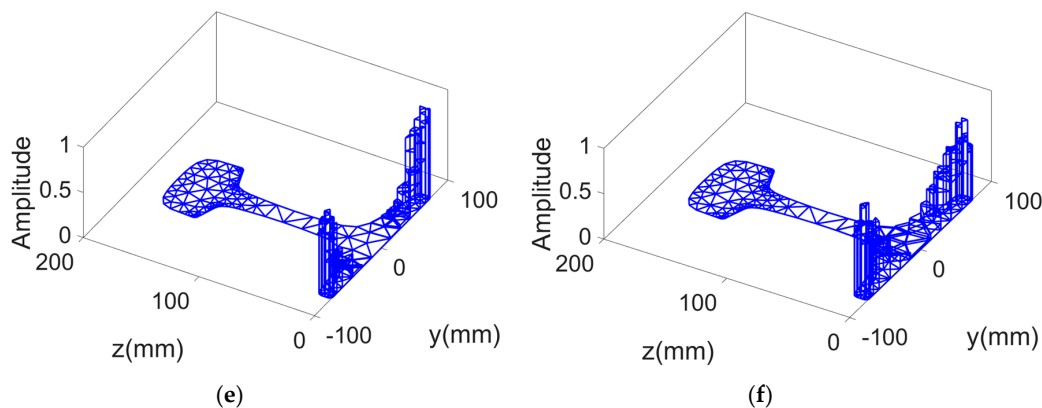

(**e**)  (**f**)

**Figure 4.** Normalized strain energy amplitude of modes. (**a**) mode 7; (**b**) mode 20; (**c**) mode 3; (**d**) mode 1; (**e**) mode 2; (**f**) mode 10.

The dispersion curves of the above modes are shown in Figure 5. When the frequency dispersion occurs, the wave packet composed of different frequency components will be dispersed, causing the waveform of the received signal to be distorted relative to the transmitted signal. Therefore, it is necessary to select modes with good non-dispersion characteristics. At the frequency of 35 kHz, the dispersion curves of mode 7 are gradual than that of mode 20. This means that the non-dispersion characteristic of mode 7 is better than that of mode 20. Hence mode 7 is selected to detect the rail head defect. Similarly, mode 3 is selected to detect the rail web defect. Mode 1, mode 2, and mode 10 are suitable to detect rail base defects.

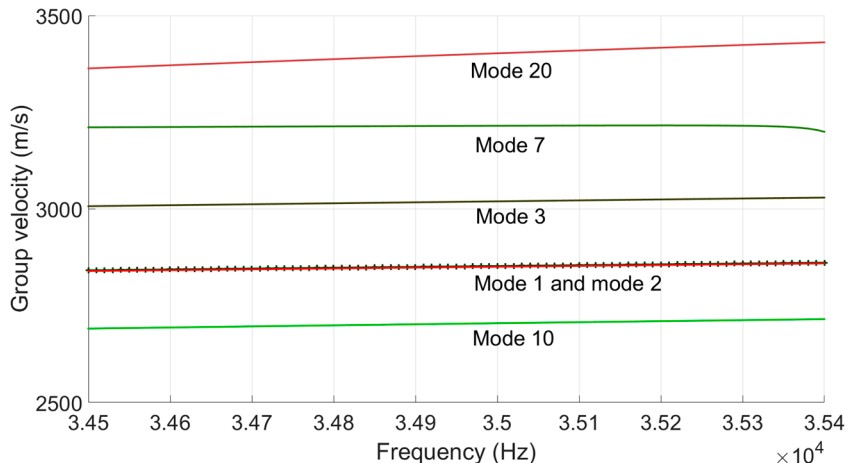

**Figure 5.** Group velocity dispersion curves.

## 3. Mode Excitation Based on Mode Matrix

In order to excite the expected modes, it is necessary to determine the optimal excitation direction and excitation node. The detailed excitation method is described in this section.

### 3.1. Determine Excitation Direction

Here Euclidean distance is used to describe the true distance between the vibration vectors of modes in the rail. The distances of displacement vector between mode $m$ and other modes in $x$, $y$ and $z$ directions are calculated, and the direction with the largest Euclidean distance is chosen as the optimum excitation direction of mode $m$.

There are $p$ nodes in the profile of rail cross section. Each node has 3 degrees of freedom. The displacements are represented as $x_i$, $y_i$, and $z_i$, respectively. The Euclidean distances $X_{mn}$, $Y_{mn}$, and $Z_{mn}$ of the vibration vectors of mode $m$ and mode $n$ in $x$, $y$, and $z$ directions can be defined as follows:

$$X_{mn} = \sqrt{\sum_{i=1}^{p}(x_{im} - x_{in})^2} \tag{4}$$

$$Y_{mn} = \sqrt{\sum_{i=1}^{p}(y_{im} - y_{in})^2} \tag{5}$$

$$Z_{mn} = \sqrt{\sum_{i=1}^{p}(z_{im} - z_{in})^2} \tag{6}$$

At the frequency of 35 kHz, there are 20 kinds of guided wave modes in rails. Assuming the expected mode is mode $m$. Firstly, we can calculate the Euclidean distance between mode $m$ and all other modes in $x$, $y$, and $z$ directions respectively at the same frequency. Then 3 matrices of $20 \times 20$ can be obtained and the column m or the row m of the matrix represents the Euclidean distance between the mode $m$ and other modes. Next, we calculate the mean of the distance between a given mode to all others in $x$, $y$, and $z$ directions and call the value $\bar{x}_m$, $\bar{y}_m$, and $\bar{z}_m$. This is done separately for all three directions. By comparing the values of $\bar{x}_m$, $\bar{y}_m$, and $\bar{z}_m$, the direction with the largest value is selected as the optimum excitation direction.

According to the above method, the mode shape data of 20 modes at the frequency of 35 kHz are substituted into Equations (4)~(6). Then 3 matrices of $20 \times 20$ can be obtained and each column or each row of a matrix represents the Euclidean distance between the corresponding mode and other modes. This paper selects mode 1 to detect the rail base defect. Then we can calculate $\bar{x}_1$, $\bar{y}_1$, and $\bar{z}_1$ of mode 1. By comparing the 3 values of $\bar{x}_1$, $\bar{y}_1$, and $\bar{z}_1$, the z direction with the largest value is selected as the optimum excitation direction. The calculation process of mode 3 and mode 7 is the same and $\bar{x}$, $\bar{y}$, and $\bar{z}$ of mode 1, mode 3, and mode 7 in $x$, $y$, and $z$ directions are shown in Table 1.

**Table 1.** The values of $\bar{x}$, $\bar{y}$, and $\bar{z}$.

| Mode | $\bar{x}$ (mm) | $\bar{y}$ (mm) | $\bar{z}$ (mm) |
|:---:|:---:|:---:|:---:|
| 1 | 0.063 | 0.054 | 0.074 |
| 3 | 0.067 | 0.077 | 0.054 |
| 7 | 0.068 | 0.053 | 0.080 |

According to the data in Table 1, the optimal excitation direction of mode 1, mode 3 and mode 7 is $z$, $y$, and $z$ direction, respectively.

### 3.2. Determine Excitation Nodes

Covariance is employed to analyze the vibration displacement deviation of each node on the rail profile from the average displacement of all nodes. Then the optimal excitation point is selected according to the overall vibration displacement of the nodes and the node with the largest covariance is selected as the optimum positive excitation node in mode $m$. If there is a negative covariance value, the node with the minimum covariance value is the best inverse excitation node. Equation (7) describes the covariance of node $t$ and node $e$ of mode $m$ in $x$, $y$, and $z$ directions at the rail base.

$$RB\_COV_m(t,e) = \frac{\sum_{s=1}^{3}(t_s - \bar{t})(e_s - \bar{e})}{3-1} \tag{7}$$

where $t_s$ is the vibration displacement of node $t$ in mode $m$. $\bar{t}$ is the average value of the vibration displacement of node $t$. $e_s$ is the vibration displacement of node $e$ of mode $m$. $\bar{e}$ is the average value of the vibration displacement of node $e$.

Similarly, Equations (8) and (9) respectively describes the covariance of node $t$ and node $e$ of mode $m$ in $x$, $y$, and $z$ directions at the rail web and rail head.

$$RW\_COV_m(t,e) = \frac{\sum_{s=1}^{3}(t_s - \bar{t})(e_s - \bar{e})}{3-1} \tag{8}$$

$$RH\_COV_m(t,e) = \frac{\sum_{s=1}^{3}(t_s - \bar{t})(e_s - \bar{e})}{3-1} \tag{9}$$

The mode shape data of mode 1, mode 3, and mode 7 is substituted into Equations (7)~(9), respectively. The optimal excitation nodes of each mode are calculated and the excitation method is selected according to mode shape characteristics. Figure 6 shows the optimal excitation nodes.

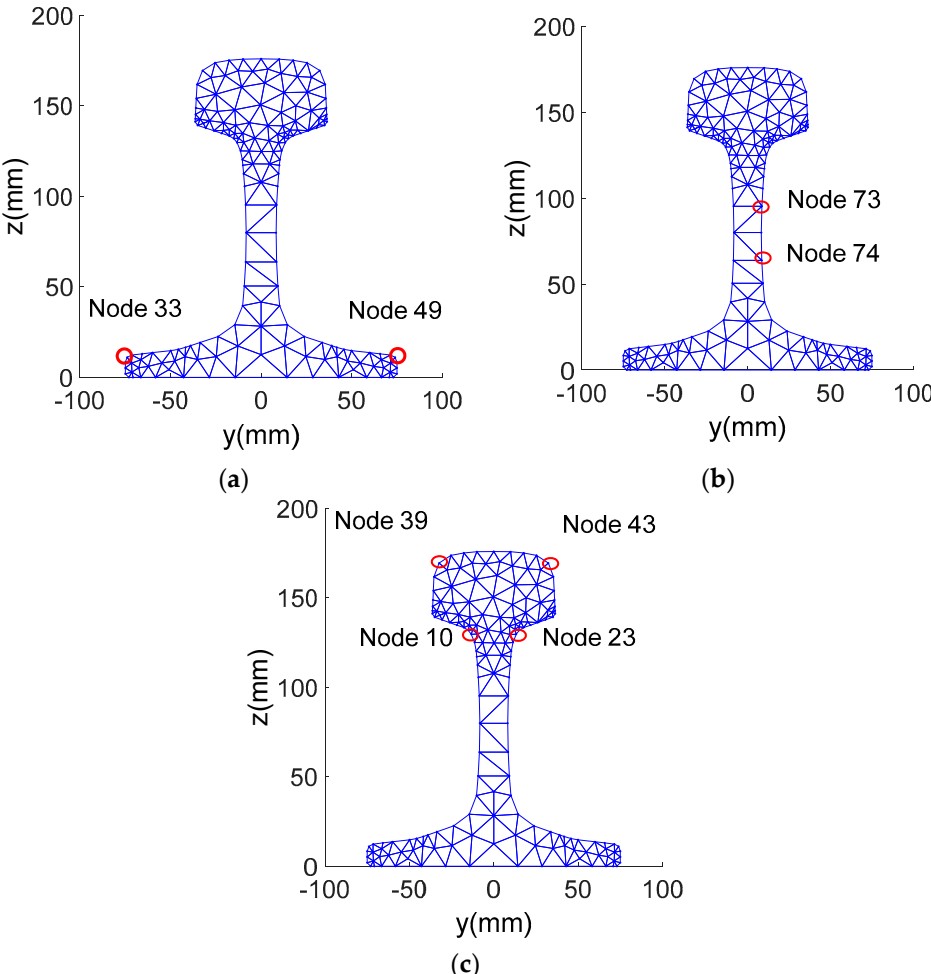

**Figure 6.** Excitation nodes of modes. (**a**) Excitation node of mode 1; (**b**) excitation nodes of mode 3; (**c**) excitation nodes of mode 7.

In summary, mode 1 selects the two-side symmetric excitation, which is excited on node 49 and node 33 in $z$ positive direction. Node 33 receives the signal in $z$ direction. Mode 3 selects the one-side symmetric excitation on node 73 and node 74 in $y$ positive direction. Node 74 is selected to receive the signal in $y$ direction. Mode 7 selects the symmetric excitation on nodes 43, node 39, node 10 and

node 23 in *z* direction. Then node 43 receives the signal in *z* direction. The detailed selection process of excitation node can be referred to our previous work [20].

## 4. Mode Excitation Simulation and Verification

Mode excitation is simulated with ANSYS to verify the above proposed method. The excitation signal is a sine wave modulated by the Hanning window with a frequency of 35 kHz as shown in Figure 7.

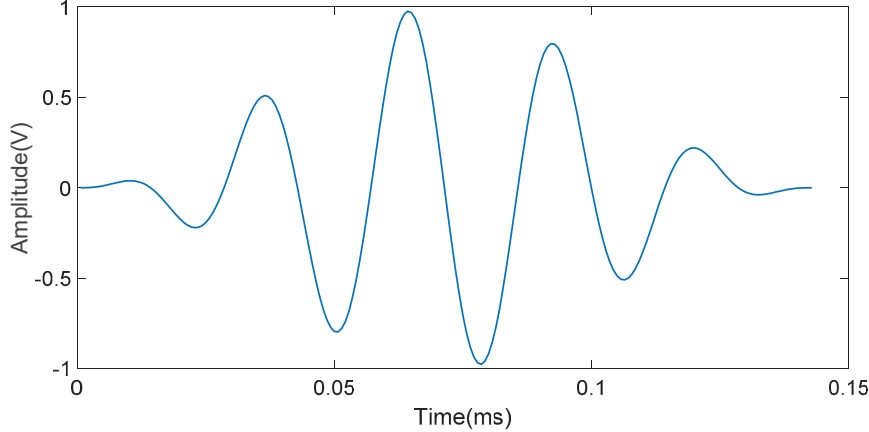

**Figure 7.** Excitation signal.

In the simulation process, the excitation signal is applied on the rail at 4 m from the right end of the rail as shown in Figure 8 in order to avoid the influence of the rail end echo on the simulation results. Between 0.8 and 2.7 m from the excitation node, a set of data acquisition array is set at intervals of 5 mm. There are 380 data acquisition nodes.

The two-dimensional Fast Fourier Transform(2D-FFT) is used on the acquired data to identify the phase velocity of the mode. The simulation results are compared with the frequency wavenumber dispersion curves calculated by SAFE method as shown in Figure 9.

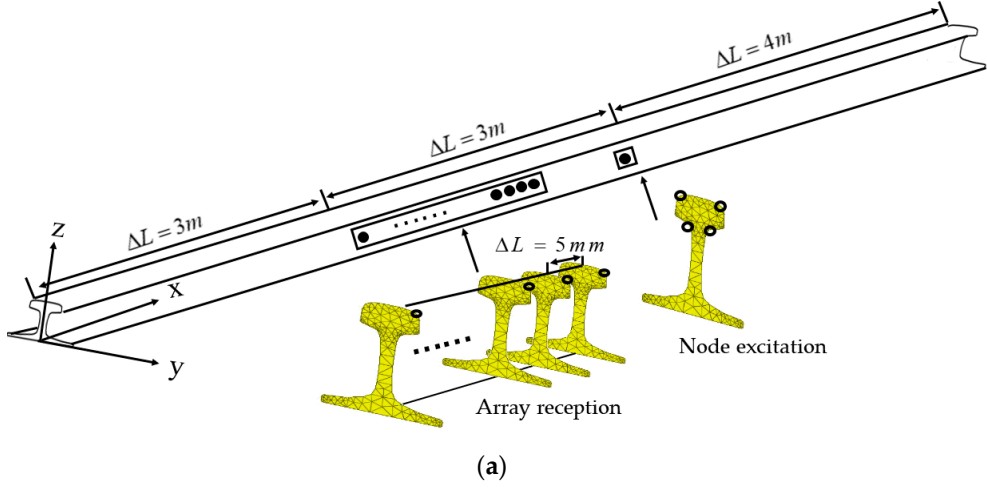

(**a**)

**Figure 8.** *Cont.*

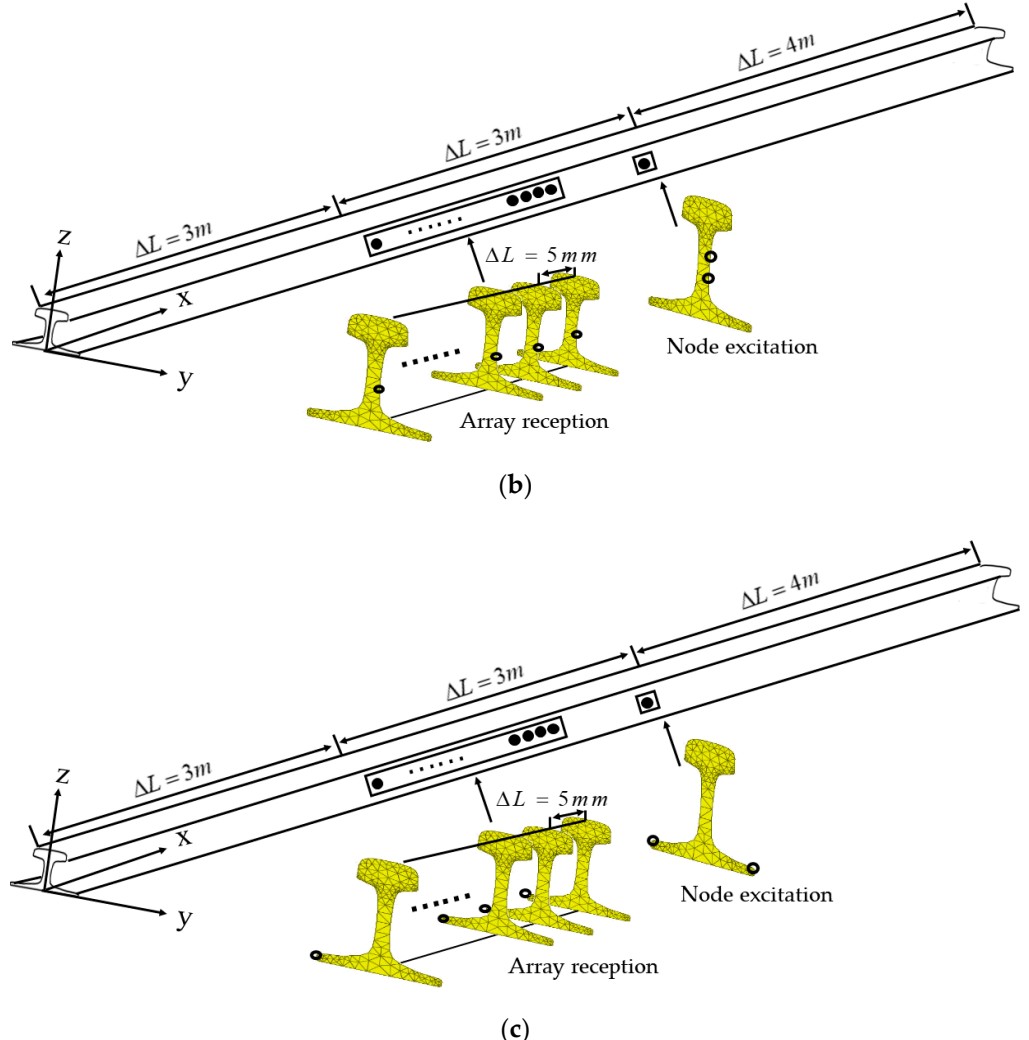

**Figure 8.** Node excitation simulation diagram. (**a**) Node excitation simulation diagram of mode 7; (**b**) node excitation simulation diagram of mode 3; (**c**) node excitation simulation diagram of mode 1.

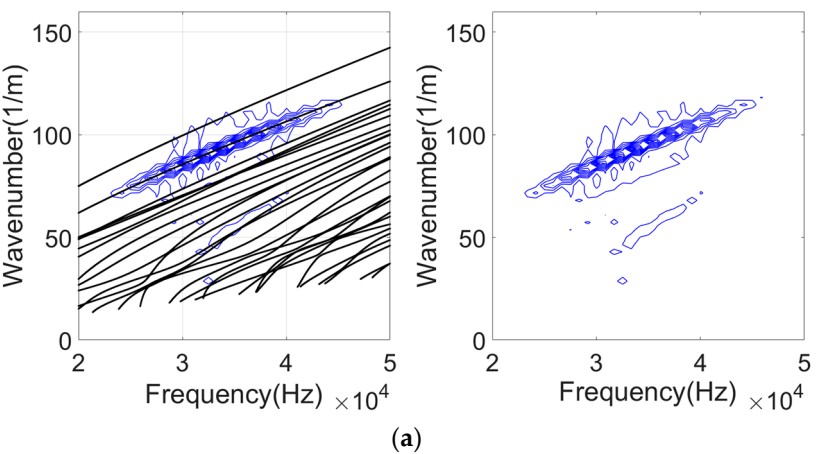

**Figure 9.** *Cont.*

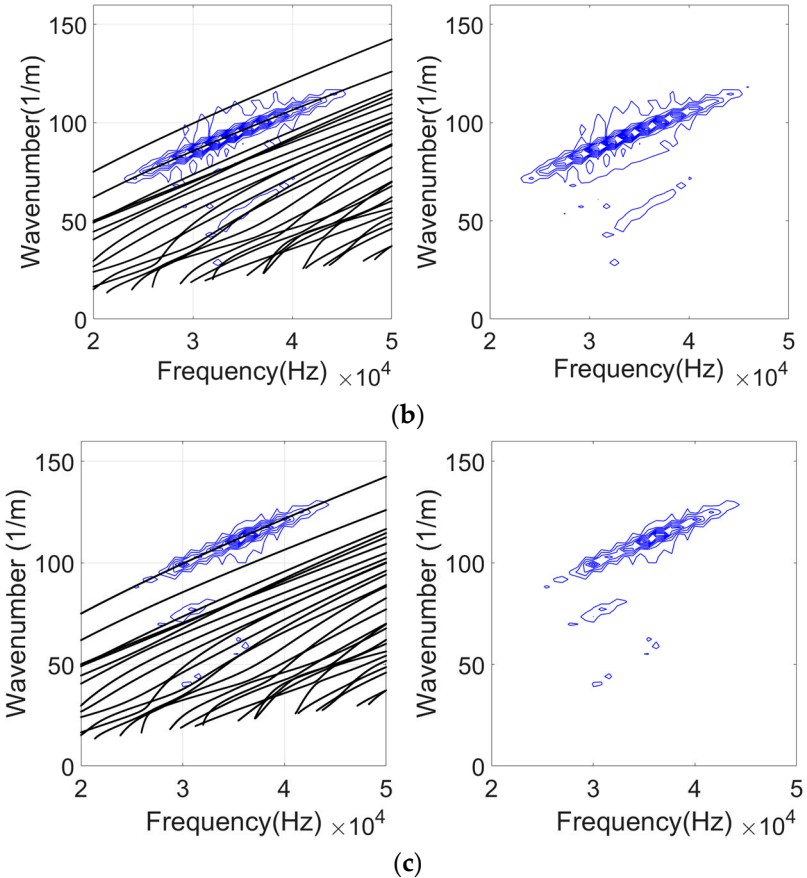

**Figure 9.** Frequency wavenumber curves. (**a**) 2D-FFT result of mode 7; (**b**) 2D-FFT result of mode 3; (**c**) 2D-FFT result of mode 1.

In Figure 9, the result of 2D-FFT coincides well with the frequency wavenumber curve of the corresponding mode. The ordinate value of the wavenumber $\xi$ can be acquired from Figure 9 when the abscissa $f$ is 35 kHz. The relationship between wavenumber $\xi$ and frequency $f$ is shown in Equation (10):

$$C_p = \frac{2\pi f}{\xi} \tag{10}$$

Then the frequency $f$ and wavenumber $\xi$ are substituted into the Equation (10) to calculate the simulation phase velocity of each mode. The calculation results are shown in Table 2.

**Table 2.** Phase velocity of modes.

| Mode | Simulated Phase Velocity Value (m/s) | Theoretical Phase Velocity Value (m/s) | Phase Velocity Error |
|---|---|---|---|
| 7 | 2669.4 | 2737.6 | 2.49% |
| 3 | 2282.5 | 2286.1 | 0.16% |
| 1 | 1982.5 | 1983.8 | 0.07% |

It is found from Table 2 that the phase velocity errors of mode 7, mode 3 and mode 1 are 2.49%, 0.16%, 0.07% respectively. It is proved that the method of node excitation is effective, but there are still some interference modes. In order to excite a relatively single mode, it is necessary to study the method of mode enhancement and interference mode suppression.

## 5. Mode Enhancement and Defect Detection Simulation

### 5.1. Mode Enhancement Theory Based on Phase Control and Time Delay Technology

According to the theory of Rose [21], the installation interval and the excitation sequence of transducer array are set based on the expected guided wave mode, and the excited guided waves have the same phase angle. The excitation time interval is set as periodic $T$. According to the principle of wave superposition, the synthetic amplitude of two coherent waves with the same phase is the largest when the interval of transducers is offered based on the expected mode wavelength. As shown in Figure 10, 5 transducers are installed on the rail.

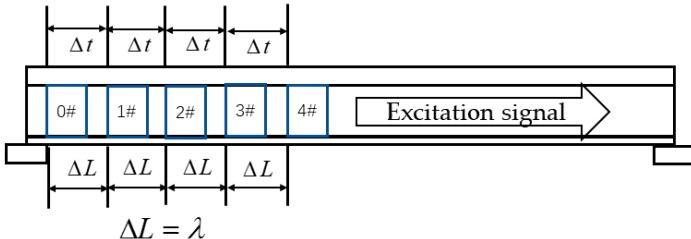

**Figure 10.** Phased array excitation diagram.

Transducer 0# is an excitation signal without time delay. The displacement function of each particle is $u_0(x, y, z, t)$.

$$u_0(x,y,z,t) = \begin{bmatrix} u_x(x,y,z,t) \\ u_y(x,y,z,t) \\ u_z(x,y,z,t) \end{bmatrix} = \begin{bmatrix} U_x(y,z) \\ U_y(y,z) \\ U_z(y,z) \end{bmatrix} e^{i(\xi x - \omega t)} \tag{11}$$

where $\xi$ is the wavenumber.

The distance between transducer 1# and transducer 0# is $\lambda$, and the time delay is $T$. The displacement function of each particle is $u_1(x, y, z, t)$.

$$u_1(x,y,z,t) = \begin{bmatrix} u_x(x,y,z,t) \\ u_y(x,y,z,t) \\ u_z(x,y,z,t) \end{bmatrix} = \begin{bmatrix} U_x(y,z) \\ U_y(y,z) \\ U_z(y,z) \end{bmatrix} e^{i[\xi(x - \Delta L) - \omega(t - \Delta t)]} = \begin{bmatrix} U_x(y,z) \\ U_y(y,z) \\ U_z(y,z) \end{bmatrix} e^{i[\xi(x - \lambda) - \omega(t - T)]} \tag{12}$$

As we know:

$$\xi\lambda = 2\pi \tag{13}$$

$$\omega T = 2\pi \tag{14}$$

Substitute the Equations (13) and (14) into the Equation (12) to obtain the Equation (15).

$$u_1(x,y,z,t) = \begin{bmatrix} u_x(x,y,z,t) \\ u_y(x,y,z,t) \\ u_z(x,y,z,t) \end{bmatrix} = \begin{bmatrix} U_x(y,z) \\ U_y(y,z) \\ U_z(y,z) \end{bmatrix} e^{i[\xi(x - \lambda) - \omega(t - T)]} = \begin{bmatrix} U_x(y,z) \\ U_y(y,z) \\ U_z(y,z) \end{bmatrix} e^{i[\xi x - \omega t]} = u_0(x,y,z,t) \tag{15}$$

Similarly, the displacement and phase of transducer array 0#, 1#, 2#, 3#, and 4# can be derived as Equation (16):

$$u_0(x,y,z,t) = u_1(x,y,z,t) = u_2(x,y,z,t) = u_3(x,y,z,t) = u_4(x,y,z,t) \tag{16}$$

When the guided wave propagates along x direction, the expected mode is enhanced and the interference mode is suppressed. The transducer intervals of each mode are shown in Table 3.

**Table 3.** Transducer intervals.

| Mode | Phase Velocity Value (m/s) | ΔL (mm) |
|------|---------------------------|---------|
| 7 | 2737.6 | 78 |
| 3 | 2286.1 | 65 |
| 1 | 1983.8 | 57 |

*5.2. Mode Enhancement Simulation Results*

The simulation process of mode enhancement is the same as Section 3, only the excitation method is changed to transducer array. The simulation results are compared with the frequency wavenumber dispersion curves as shown in Figure 11.

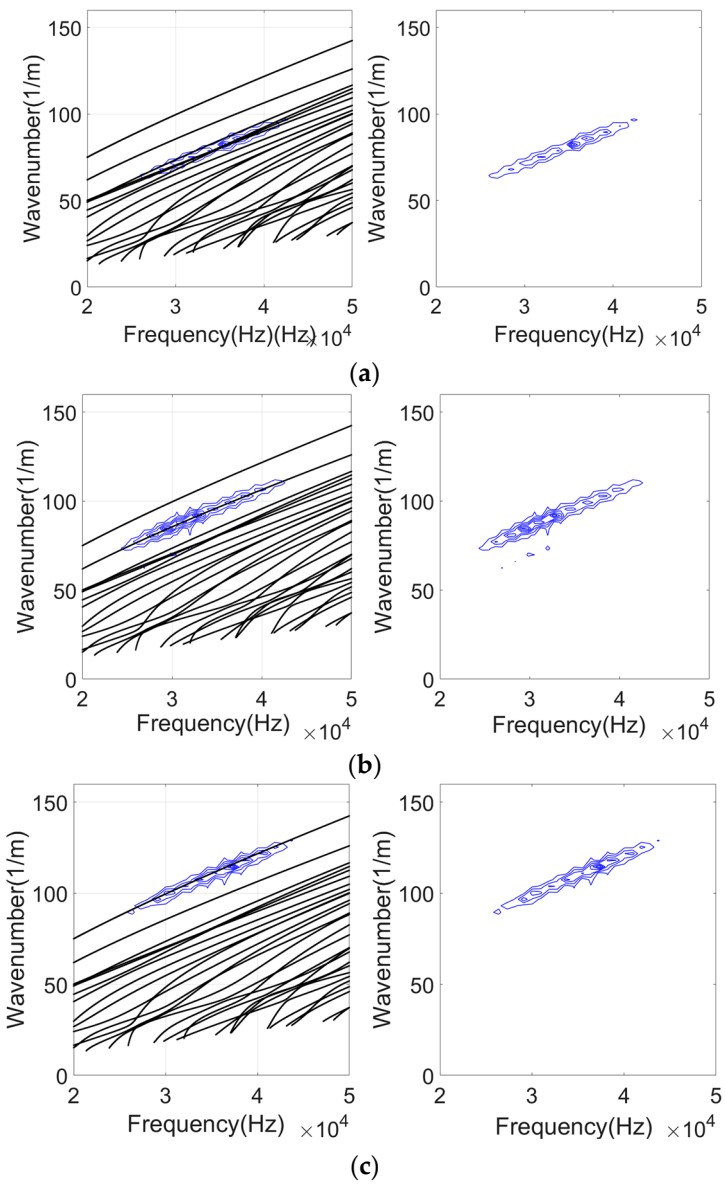

**Figure 11.** Frequency wavenumber curves. (**a**) 2D-FFT result of mode 7; (**b**) 2D-FFT result of mode 3; (**c**) 2D-FFT result of mode 1.

From Figure 11 we can see, the result of 2D-FFT coincides well with the frequency wavenumber curve of the corresponding mode and there is no interference mode. The wavenumber $\xi$ and frequency

*f* are substituted into Equation (10). The calculation results of simulated phase velocities are shown in Table 4, and compared with theoretical values. It can be seen that the maximum error is 0.68%.

**Table 4.** Phase velocities of modes.

| Mode | Simulated Phase Velocity Value (m/s) | Theoretical Phase Velocity Value (m/s) | Phase Velocity Error |
|------|------|------|------|
| 7 | 2719.0 | 2737.6 | 0.68% |
| 3 | 2281.5 | 2286.1 | 0.20% |
| 1 | 1982.0 | 1983.8 | 0.09% |

The above discussion is about phase velocity, then the process of the group velocity calculation is described as following. The simulation data at 1 m and 2.2 m from the excitation nodes are extracted and the simulation results of mode 7, mode 3, and mode 1 are shown in Figure 12.

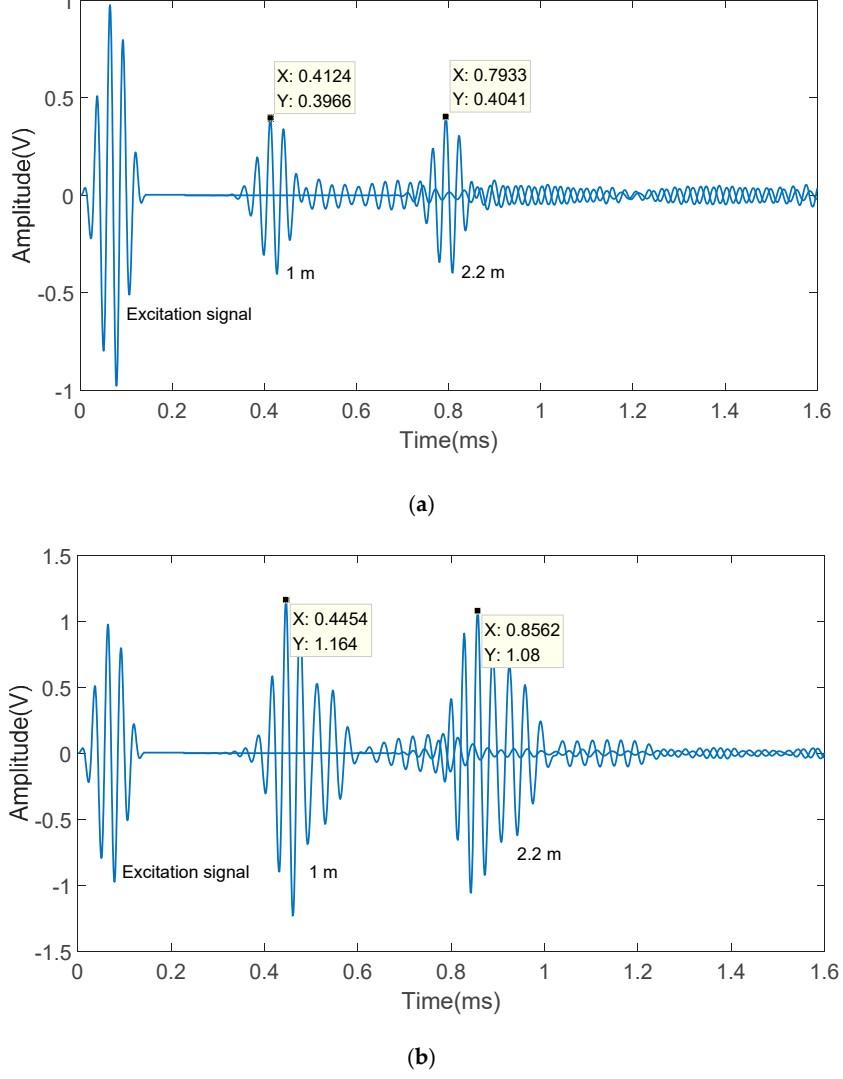

(a)

(b)

**Figure 12.** *Cont*.

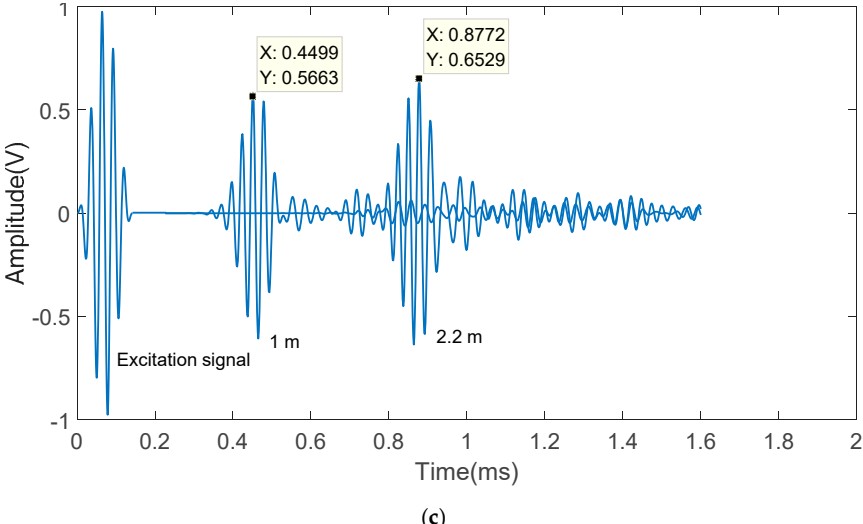

(**c**)

**Figure 12.** Mode simulation results. (**a**) Mode 7 simulation result; (**b**) Mode 3 simulation result; (**c**) Mode 1 simulation result.

The simulation group velocity can be calculated by calculating the peak time difference between 1 m and 2.2 m and the results are shown in Table 5.

**Table 5.** Group velocities of modes.

| Mode | Simulated Group Velocity Value (m/s) | Theoretical Group Velocity Value (m/s) | Group Velocity Error |
|---|---|---|---|
| 7 | 3150.4 | 3215.1 | 2.01% |
| 3 | 2994.0 | 3019.9 | 0.86% |
| 1 | 2818.2 | 2852.7 | 1.21% |

As can be seen from Table 5, the group velocity error of each mode is relatively small. The results show that the method of array excitation can enhance the expected mode and suppress the interference mode, which indicates a relatively single mode is successfully excited.

*5.3. Defect Detection Simulation*

At present, rail head transverse defects are the most dangerous damage for rail operation. Figure 13a presents the simulation of detecting an internal transverse defect. The shape of the defect on the rail cross section is a circle with a diameter of 23 mm and a length of 5 mm in x direction. In Figure 13b, the rail web defect is assumed as a vertical crack. The depth of the crack in z direction is 30 mm and 5 mm in x direction. In Figure 13c, the rail base defect is a transverse defect, the length of which is 25 mm in the y direction and 10 mm in x direction. In defect detection simulation, the modes excitation and data acquisition method are the same as Section 5.2 and the rail defects are 3 m from the left end of the rail.

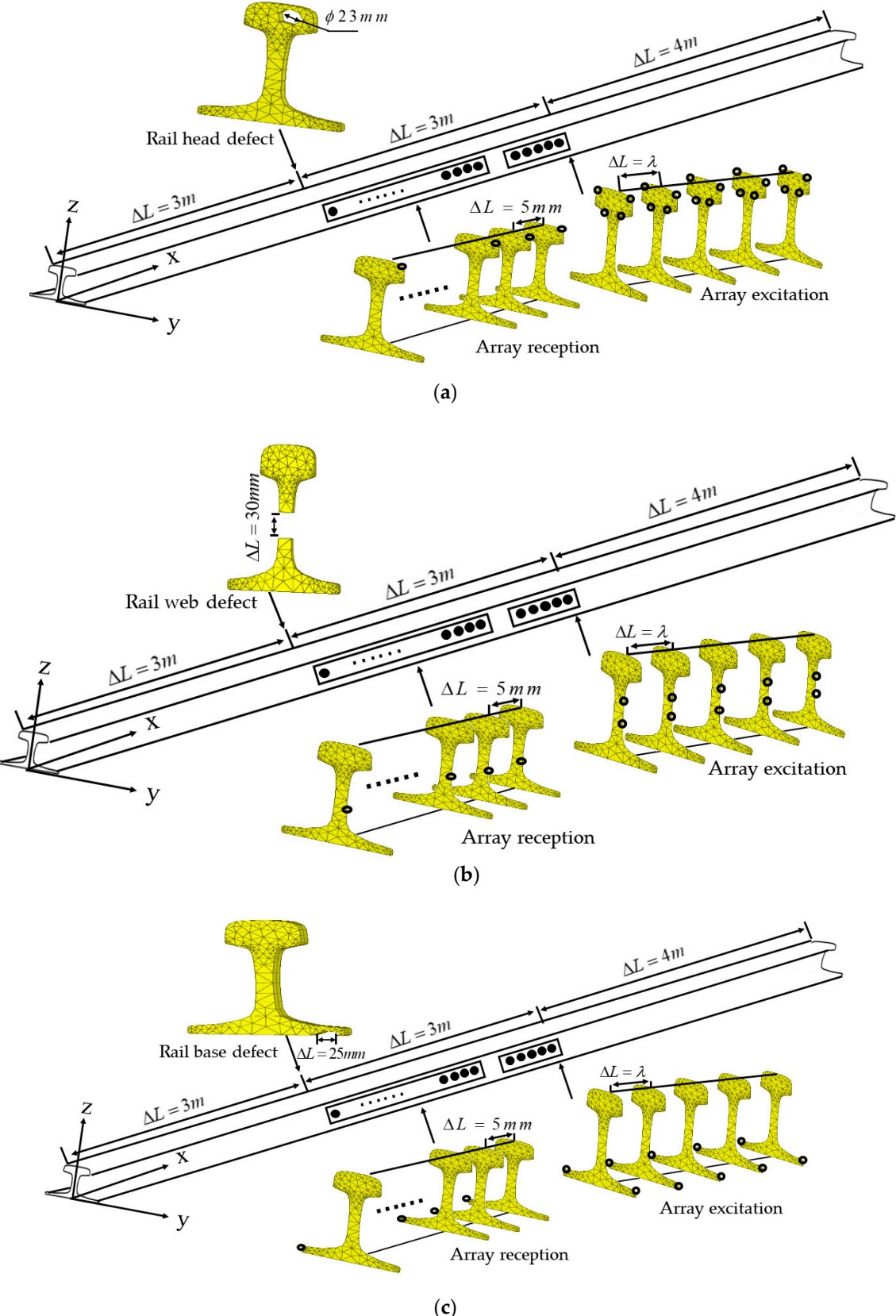

**Figure 13.** Defect detection simulation diagram. (**a**) diagram of array excitation mode 7 defect detection; (**b**) diagram of array excitation mode 3 defect detection; (**c**) diagram of array excitation mode 1 defect detection.

It is necessary to verify whether the received defect echo at the receiving nodes has occurred in mode conversion before calculating the defect location. The defect echoes extracted separately are transformed by 2D-FFT as shown in Figure 14.

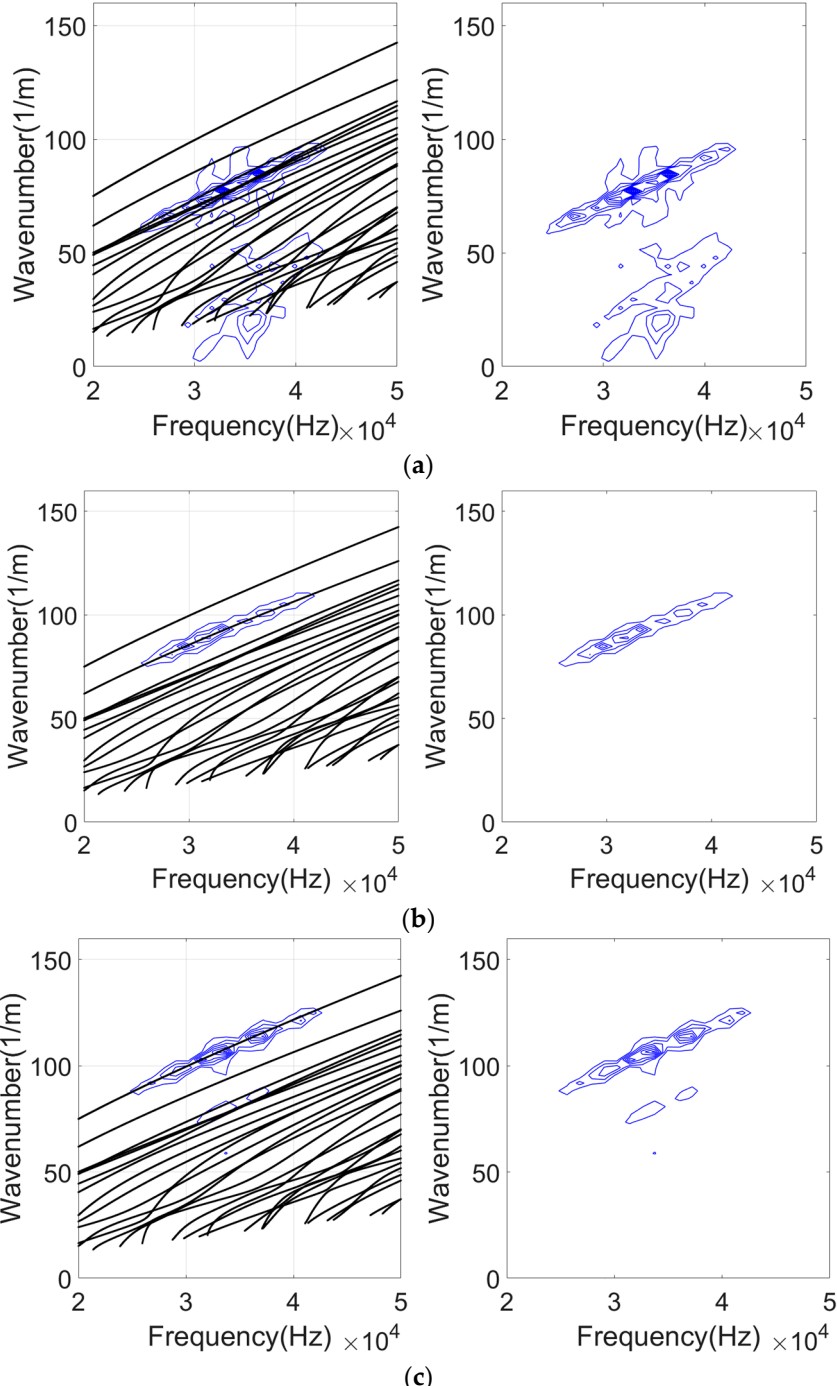

**Figure 14.** Frequency wavenumber curves. (**a**) mode 1 defect reflected echo 2D-FFT results; (**b**) mode 3 defect reflected echo 2D-FFT results; (**c**) mode 7 defect reflected echo 2D-FFT results.

Similarly, the phase velocity of each mode can be calculated from the frequency wavenumber curve. Table 6 shows the phase velocities of reflected echo modes.

**Table 6.** Phase velocities of reflected echo modes.

| Mode | Simulated Phase Velocity Value (m/s) | Theoretical Phase Velocity Value (m/s) | Phase Velocity Error |
|---|---|---|---|
| 7 | 2721.0 | 2737.6 | 0.61% |
| 3 | 2262.5 | 2286.1 | 1.03% |
| 1 | 1994.6 | 1983.8 | 0.54% |

As can be seen from Table 6 that the error between the simulated phase velocity and the theoretical phase velocity of each defect echo can be accepted for application. The reflected echoes received at the corresponding nodes have not occurred mode conversion.

During the process of simulation, the receiving node is 1 m away from the defect. The position of the rail defect can be obtained by calculating the time difference between the received excitation signal and the reflected echo at the receiving node. As shown in Figure 15, the defect is 3 m from the left end of the rail. The receiving node is 1 m from the defect and 2 m from the excitation signal.

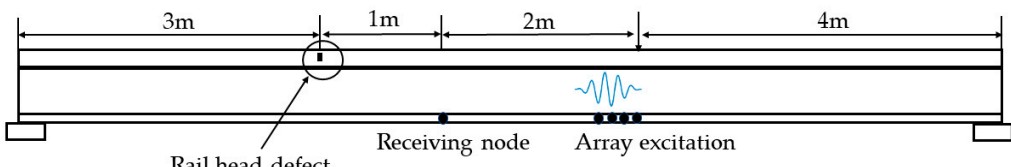

**Figure 15.** Diagram of the rail head defect detection.

When the defect is in the rail head, the detection results of mode 7, mode 3 and mode 1 are as shown in Figure 16. Only mode 7 has reflected echo. The time difference between the received excitation signal and the reflected echo is 0.6137 ms.

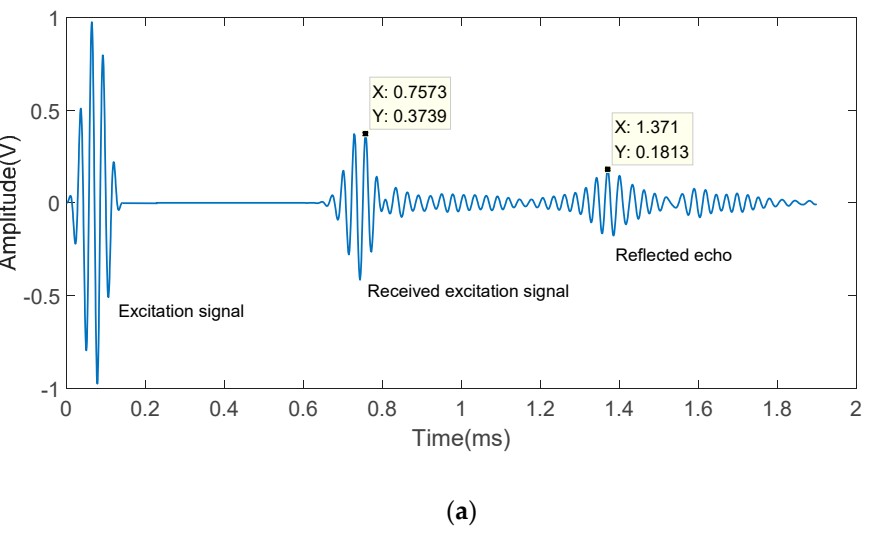

(**a**)

**Figure 16.** *Cont.*

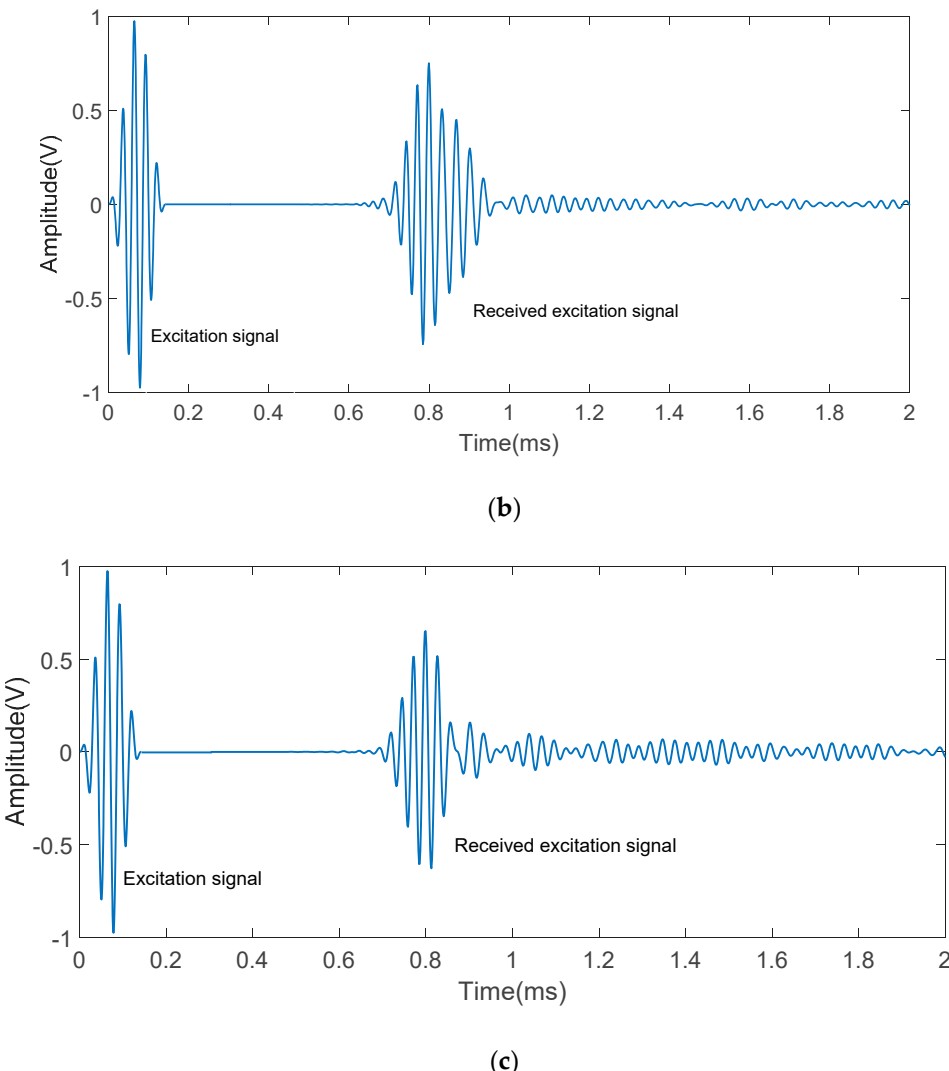

(**b**)

(**c**)

**Figure 16.** Rail head defect detection. (**a**) Mode 7 rail head defect detection; (**b**) mode 3 rail head defect detection; (**c**) mode 1 rail head defect detection.

When the defect is in the rail web, only mode 3 has reflected echo as shown in Figure 17 and the time difference is 0.6478 ms.

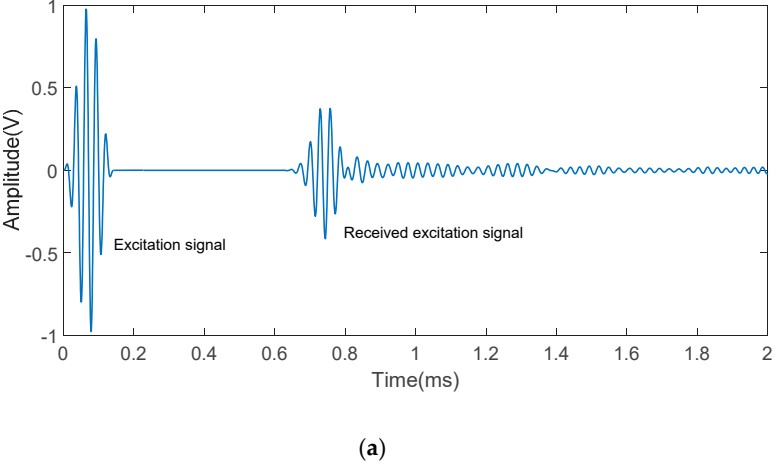

(**a**)

**Figure 17.** *Cont.*

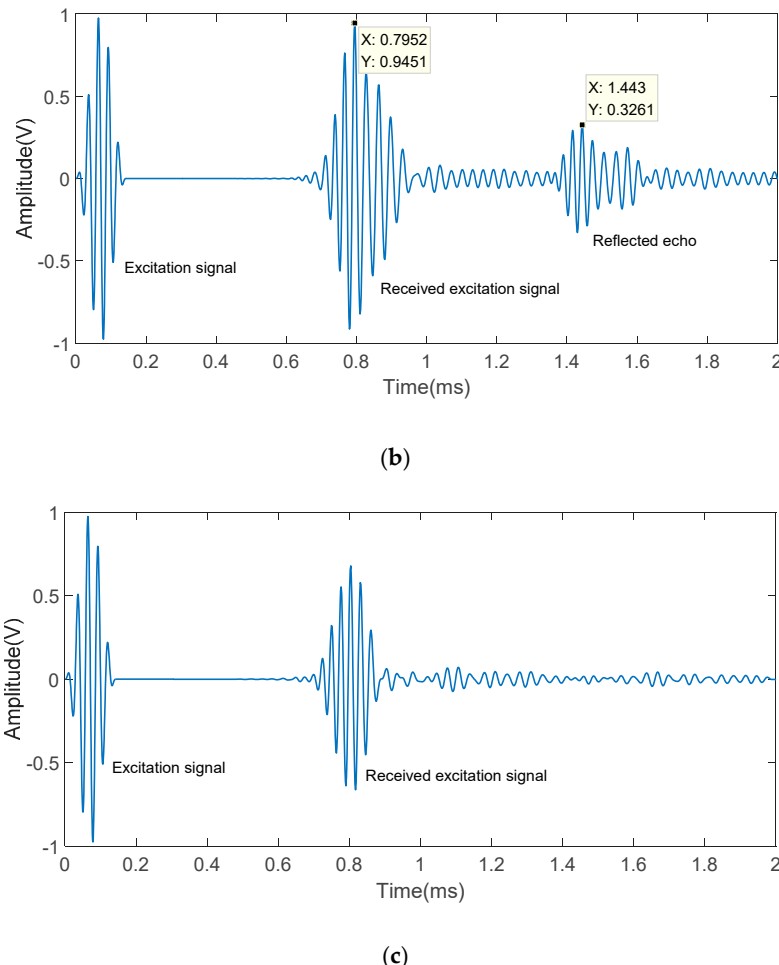

(b)

(c)

**Figure 17.** Rail web defect detection. (**a**) Mode 7 rail web defect detection; (**b**) mode 3 rail web defect detection; (**c**) mode 1 rail web defect detection.

Similarly, when the defect is in the rail base, only mode 1 has reflected echo as shown in Figure 18 and the time difference between is 0.6913 ms.

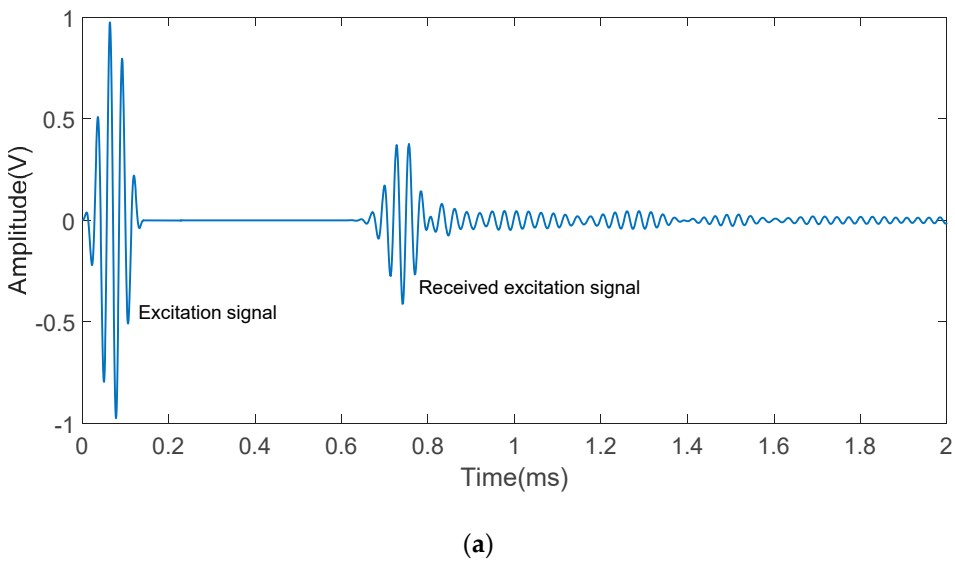

(a)

**Figure 18.** *Cont.*

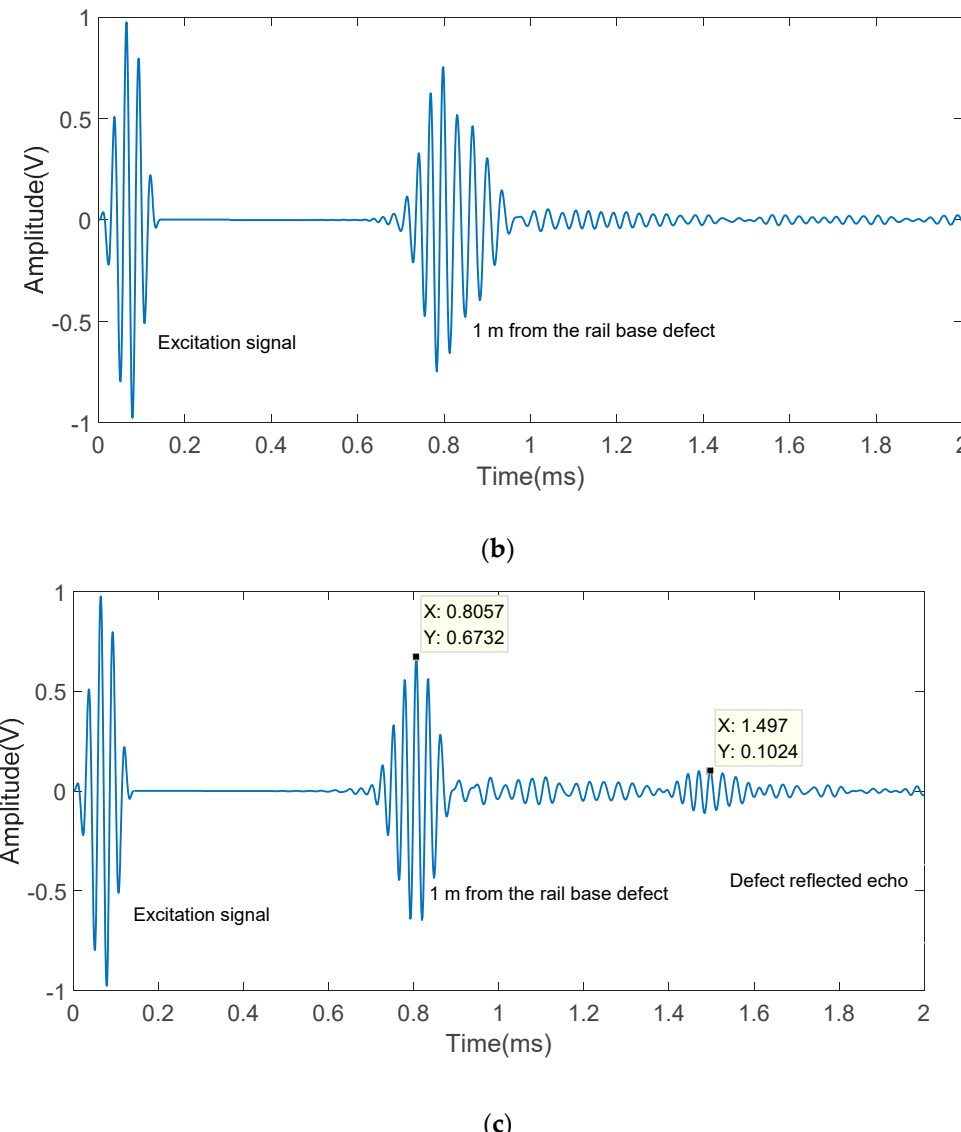

**Figure 18.** Rail base defect detection. (**a**) Mode 7 rail base defect detection; (**b**) mode 3 rail base defect detection; (**c**) mode 1 rail base defect detection.

The defect position can be calculated based on the time difference. Table 7 shows the detect results of the defect position. The results can be seen from the Table 7 that the defect positioning error of mode 7, mode 3 and mode 1 is 1.34%, 2.19%, and 1.40% respectively.

**Table 7.** Defect detection of each mode.

| Mode | Time Difference (ms) | Theoretical Group Velocity Value (m/s) | Simulation Defect Position (m) | Positioning Error |
|------|------|------|------|------|
| 7 | 0.6137 | 3215.1 | 0.9866 | 1.34% |
| 3 | 0.6478 | 3019.9 | 0.9781 | 2.19% |
| 1 | 0.6913 | 2852.7 | 0.9860 | 1.40% |

## 6. Conclusions

In this paper, a novel method for mode selection and excitation in rail defect detection is proposed based on ultrasonic guided waves. The mode shape data in the CHN60 rail is obtained at the frequency of 35 kHz by using SAFE method. And the modes with good non-dispersion characteristics, which

are respectively sensitive to the defects of the rail head, rail web and rail base, are selected combining the strain energy distribution diagrams with group velocity dispersion curves. The optimal excitation direction of the expected modes is solved by the Euclidean distance, and the optimal excitation node is calculated by covariance matrix. Then, phase control and time delay technology is applied to achieve the enhancement of expected modes and suppression of interferential modes. After that, ANSYS is used to simulate the expected modes excitation and defects detection in the different rail parts with phase velocity and group velocity to validate the proposed methods. The simulation results present an acceptable error and demonstrate the effectiveness of the guided wave mode selection and excitation method, which proposes a feasible scheme for the field test and application to rail defect detection.

**Author Contributions:** This paper is a result of the full collaboration of all the authors; H.S. and L.Z. (Lu Zhuang) conceived and proposed the main idea of the paper; X.X. combined with phase control and time delay technology to improve the mode excitation method; L.Z. (Liang Zhu) simulated the defect detection and analyzed the data; L.Z. (Liang Zhu) and Z.Y. put forward their views on data processing; H.S. and L.Z. (Lu Zhuang) wrote the paper.

**Acknowledgments:** The work is supported by the National Key Research and Development Program of China (2016YFB1200401), Supported by Foundation of Key Laboratory of Vehicle Advanced Manufacturing, Measuring, and Control Technology (Beijing Jiaotong University), Ministry of Education, China.

**Conflicts of Interest:** The authors declare no conflict of interest.

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
