# Peer review of "An Ultrasonic Guided Wave Mode Selection and Excitation Method in Rail Defect Detection"

_applsci, doi:10.3390/app9061170_

Round 1
Reviewer 1 Report
I believe this article is suitable for your journal.
Author Response
Dear Professor:
Thank you for your comments on our paper entitled “An Ultrasonic Guided Wave Mode Selection and Excitation Method in Rail Defect Detection” (applsci-424469). We have revised our manuscript. We sincerely hope this manuscript will be finally acceptable to be published on Applied Sciences. Revised portion are marked in Red.

Reviewer 2 Report
The same group has published similar work, as given in the publication as Ref. 20: Xu, X.N.; Zhuang, L.; Xing, B.; Yu, Z.J.; Zhu, L.Q. An Ultrasonic Guided Wave Mode Excitation Method in Rails[J]. IEEE Access. 2018,6: 60414-60428. The authors have to give clear evidence that they have already worked in this field and have to give clear statements about the novelty of the present work. What is new and goes beyond Ref. 20.
0. Introduction: The authors have to mention already in the introduction their similar work, given in Ref. 20. And they have to state clearly what this new work goes beyond Ref. 20
p 2, line 73: Give a reference that explains what CHN60 means
p 2, Fig. 1: This is not a right hand system. Please correct. Make it consistent with Fig. 10
p 3 and 4: Delete Figs. 3 and 4 and Table 1: The indentical data is already published in Ref. 20. The reviewer finds that the information is not essential for the current publication
Author Response
Dear Professor:
Thank you for your comments on our paper entitled “An Ultrasonic Guided Wave Mode Selection and Excitation Method in Rail Defect Detection” (applsci-424469). We have revised our manuscript according to your kind advices and detailed suggestions. We sincerely hope this manuscript will be finally acceptable to be published on Applied Sciences. Revised portion are marked in Red. The responses to your comments are as follows:
1) Introduction: The authors have to mention already in the introduction their similar work, given in Ref. 20. And they have to state clearly what this new work goes beyond Ref. 20
Response: Thanks for your kind comments. Our team has been working on ultrasonic guided waves in rails for a long time. The two papers focus on two different contents. Ref. 20 focuses on excitation method, and this paper emphasizes on rail defect detection based on ultrasonic guided waves. In Ref. 20, an excitation method for guided wave modes in rails is proposed. And the excitation response method is used to verify the results by measuring the group velocity of guided wave mode. The research work beyond Ref. 20 are as follows:
(1) The mode selection method is studied in this paper. Combining the strain energy distribution diagrams and phase velocity dispersion curves, the mode with better non-dispersive characteristics and energy concentrated in a single part of the rail is selected for rail defect detection. The results show that the selected specific modes are sensitive to the defects of different position and the positioning error is small enough for the maintenance staff to accept.
(2) Moreover, the guided wave mode excitation method is improved based on the Ref. 20. Phase control and time delay technology is employed to achieve the expected modes enhancement and interferential modes suppression. The results show that we can excite a relatively single expected mode.
(3) The different simulation method is used in this paper. We have carried out three-dimensional simulation by using ANSYS to simulate rail modeling and defect detection.
(4) Mode validation methods are different. In this paper, we measure the group velocities and phase velocities of the expected modes. By comparing the results of 2D-FFT transform with the frequency-wavenumber curves, it is further confirmed that the relatively single expected mode is successfully excited.
2) p 2, line 73: Give a reference that explains what CHN60 means
Response: Thanks for your suggestions. We have explained it in the paper in line 74. CHN60 is a 60 kg/m China rail.
3) Fig. 1: This is not a right hand system. Please correct. Make it consistent with Fig. 10
Response: Thanks for kind suggestions. The coordinate system of the paper is not a right hand system. All the coordinate systems of the paper are consistent with Figure 1. The coordinate system of Figure 10 and Figure 15 has been corrected.
4) p 3 and 4: Delete Figs. 3 and 4 and Table 1: The identical data is already published in Ref. 20. The reviewer finds that the information is not essential for the current publication.
Response: Thanks for your kind comments. We have deleted Figure 3, Figure 5 and Table 4 from the paper.
Reviewer 3 Report
The manuscript of this first revision is considerably improved compared to the original submission. Nevertheless, there are still shortcomings which does not allow to publish the paper as it is.
The comments are given in the appended pdf as respones to your “authors response to reviewer 4”

Author Response
Dear Professor:
Thank you for your comments on our paper entitled “An Ultrasonic Guided Wave Mode Selection and Excitation Method in Rail Defect Detection” (applsci-424469). We have revised our manuscript according to your kind advices and detailed suggestions. We sincerely hope this manuscript will be finally acceptable to be published on Applied Sciences. Revised portion are marked in Red. The responses to your comments are as follows:
1) I did understand you. From your manuscript I see, that you took the amplitude of the displacement vector as measure of the “vibration energy” and say, that this is a measure of the strength of the echo generated when this wave hits a defect like a crack. I do not see, that this can be true in general! From where do you take this argument? Is there some publication claiming that? If yes, please cite it. If you found that the first time, then prove it!
Contrary to your assumption, I can image a mode where part of the rail e.g. the head is moving with a large displacement amplitude but with a very small strain in the head itself. Than the stresses of the wave in that part are also small. Further, if there are no stresses the wave does not feel a defect like a crack (the crack does introduce stress free boundary condition at crack faces and that is why the crack scatters the wave). If the boundary conditions (no normal stress) are fulfilled by the wave already ( e.g. because the wave does not have any stresses in that region) the wave is not influenced by the defect et al. It would be much better to take the amplitude of the stress in a given area as the measure of the sensitivity of that wave to defects in the discussed area. The situation would be different if inclusion like defects with (ideally) the same stiffness but much different density would be relevant. As far as I see, inclusions are not the relevant defects in the case of maintenance you are interested in.
Response: Thanks for your significant comments and suggestions. After thorough discussions, we realize that the previous idea of measuring energy by vibration displacement is wrong. We agree with your opinions and really appreciate your comments. Following your comments, we changed the method of assessing energy. The strain energy of each triangular element is calculated and the normalized strain energy distribution diagram is plotted. The modes with strain energy concentrated in a single part of the rail are selected according to the strain energy of the modes. The related content has been modified in the Section 1.
2) Your comment to me is much clearer as the text in your (revised) manuscript. So, please use a more precise and explicit language. The distance is defined between two modes. Of course, you can average the distance of a given mode to all others. But that is not the distance of the given mode as you write in some places!
You write:
“Next, the columns of the 3 matrices are averaged to acquire the average values of Euclidean distance for each mode in x, y, and z directions. By comparing the 3 values, the direction with the largest value is selected as the optimum excitation direction”. Which three values do you mean? The reader has to guess that you talk about the values for a given (selected) mode. However, because you do not mention that, it is still confusing.
What about: “Next, the columns of the matrices are averaged to acquire the average values of Euclidean distance for each mode to all others. This is done separately for all three directions.
Perhaps it is even better to introduce a new word for the mean distance of a given mode form all others. Kind of “we calculate the mean of the distance between a given mode to all others and call the value xxxxxx”
I try to phrase you argumentation for selecting the excitation direction with my words:
You define a distance between two given modes for each of three Caresian directions (equ. (4) to (6)). So Xmn is a distance between the modes m and n, where in the definition only the x displacements are involved.
Response: Thanks for your kind comments. According to your comments, we modified the corresponding content in Section 2 and improved language expression. xm, ym,zm and for the mean distance of mode m from all others are introduced.
3) You selected interesting modes by other arguments. As excitation direction for the modes you allow only the three orthogonal coordinate directions, that is you do not assume directions which are not parallel to the coordinate axis! Your claim is: a given mode is excited best in that direction where the distance (as defined before) between the selected mode and all others is largest.
Where does this claim come from? It is not intuitive for me. To construct a counter example: All modes but one have a large displacement amplitude in x direction. Unfortunately, you have selected the mode with the small displacement amplitude in x. The average distance of that mode to all others will be large! So, you would choose it. For me, intuitively, it would be a bad choice. Would it not be better to choose a criterion based on the mode itself, as it would be the average displacement amplitude in the given direction?
Response: Thanks for your kind comments. It’s true that the excitation direction is limited in x, y and z directions in our present research. Just as you said, x, y, z may not be the optimum excitation direction. But in order to simplify the problem and make it easy to be applied in the practical engineering, we choose the direction in x, y, z, not considering the special excitation angle, which need customize the transducers. As for the counter example you gave, it is right that our algorithm doesn’t work in that situation. When the excitation frequency changes from low to high, the modes become more and more complicated. From the simulation results, that situation does not exist and we have not considered this before. In our future research this is an improve direction to optimize the algorithm.
4) By the way: your definition for the distance depend on the number p, which depends on your grid and not alone on the mode. That is why you get so large numbers in Table 2. It would be better to normalize.
Response: Thanks for your kind suggestions. We have processed the data in Table 2.

Round 2
Reviewer 3 Report
I am glad you followed most of the recommendations. I recommend now: "accept"
This manuscript is a resubmission of an earlier submission. The following is a list of the peer review reports and author responses from that submission.
Round 1
Reviewer 1 Report
see attached file

Reviewer 2 Report
I was impressed with this paper.
The merit of this paper,
(1)The paper made clear the relationship between the guide wave mode and its amplication distribution.
(2)The paper made clear the optimal drive position of the specific guide wave.
(3)The paper made clear the drive condition to excite the specific guide wave.
(4)The paper shows the received signal by any defects.
I think the paper has enough qualification to be the paper of this Journal.
Fine indications are shown below,
A. Subtitle of each figure of Fig.5 is needed.
B. Fig.12 and Fig.20 is difficult for me to understand.
C. The characters in Fig.18, Fig.22, Fig.23, Fig.24, Fig.25, Fig.26, Fig.27 are too small.
D. The detail explanation in Fig.18, Fig.22, Fig.23, Fig.24, Fig.25, Fig.26, Fig.27 is needed.
Reviewer 3 Report
In this paper an interesting method is proposed and discussed for mode selection and excitation of ultrasonic guided waves in rails. There are modes which are concentrated in the head, the web and the base of the rail, respectively. The authors compare point excitation with ANSYS simulation with SAFE calculated dispersion relations. They find good agreement. An improvement is made with phased array excitation. The method is applied successfully for defect detection. However, the authors don't give dimensions for the defects.
Give less digits for velocities, 5 siginificant digits are sufficient
Fig. 1: xyz: this is not a right hand system
Fig. 4-8, 10, 11: give units (e.g. in the figure captions)
Table 2: give units
Fig. 10: give less digits for cp, 5 significant digits are sufficient
Fig. 12, 20: xyz coordinate system is too small. The excitation points are hard to be seen. Please mark them more prominent.
Fig. 19: please mark more clearly where is the defect situated and where are the reception transducers
Fig. 22-27: The amplitude values should be given relative to the excitation amplitude
p 1, line 29: write what CWR means
p 1, line 41 and p 2, line 52: give the size of the defect that was detected
p 2, line 52: write "Long and Loveday" for Ref 14
p 8, line 206: give the exact excitation function (gaussian shaped harmonic?)
p 13, line 306: give the exact dimensions of the defects
Reviewer 4 Report
The material the manuscript is based on, seems to be solid. Unfortunately, it is hard to follow the arguments of the authors in some central points of the manuscript. That concerns the selection of the modes to be used, the excitation direction and the points on the rail where these modes should be excited.
In line 114 you speak about the "vibration displacement energy". Implicitly you say that the relevant mode should have a large displacement amplitude in the area of the rail, where the defect is expected. But when a mode has a large HOMOGENEOUS displacement, that is there is no stress in that area, it will NOT be influenced by a crack. So, your argument is not valid. In that case you will not see an influence of a defect.
The procedure in chapter 3.1. is unclear to the reviewer. The rhs of (4), (5) and (6) can be calculated. The lhs of (4) is an X (without index) multiplied with a rho_nm (with index). How do make the factorisation? Surprisingly in Table 2 the X depends on the mode (so must have an index?). That is all unclear.
The same holds for chapter 3.2. Why is some covariance relevant to the selection of excitation point?
I recommend to reject the manuscript and give the authors the chance to resubmit an significantly improved manuscript.
There are some other (minor points) which should be considered when the authors will resubmit an improved manuscript:
the abbreviation SAFE was introduced but only in the abstract. Introduce any abbreviation it appears the first time in the main text.
line 94 -95 this is not a complete sentence.
line 98 put always a space between figures and the unit
fig. 4 the graphs are not readable, the units at the axis are not clear.
Table 1,3,5,7 put the units in the headline of the columns e.g. “phase velocity /(m/s)”
Fig 22 and other figures, write time/ms instead of putting 10-3 on the axis;
moreover: the graphs should be placed one below the other without a shift to the left as in you did in (c)
Table 8: why do you give the time in s? Use ms, it is easier to read. I know it is coming out that way in some software. However, you should better redraw.
